# Omeprazole Suppresses Oxaliplatin-Induced Peripheral Neuropathy in a Rodent Model and Clinical Database

**DOI:** 10.3390/ijms23168859

**Published:** 2022-08-09

**Authors:** Keisuke Mine, Takehiro Kawashiri, Mizuki Inoue, Daisuke Kobayashi, Kohei Mori, Shiori Hiromoto, Hibiki Kudamatsu, Mayako Uchida, Nobuaki Egashira, Satoru Koyanagi, Shigehiro Ohdo, Takao Shimazoe

**Affiliations:** 1Department of Clinical Pharmacy and Pharmaceutical Care, Graduate School of Pharmaceutical Sciences, Kyushu University, Fukuoka 812-8582, Japan; 2Department of Local Healthcare Science, Faculty of Pharmaceutical Sciences, Kyushu University, Fukuoka 812-8582, Japan; 3Department of Education and Research Center for Pharmacy Practice, Faculty of Pharmaceutical Sciences, Doshisha Women’s College of Liberal Arts, Kyotanabe 610-0395, Japan; 4Department of Pharmacy, Kyushu University Hospital, Fukuoka 812-8582, Japan; 5Department of Pharmaceutics, Faculty of Pharmaceutical Sciences, Kyushu University, Fukuoka 812-8582, Japan

**Keywords:** oxaliplatin, peripheral neuropathy, omeprazole, proton pump inhibitors, chemotherapy

## Abstract

(1) Background: Oxaliplatin is used as first-line chemotherapy not only for colorectal cancer but also for gastric and pancreatic cancers. However, it induces peripheral neuropathy with high frequency as an adverse event, and there is no effective preventive or therapeutic method. (2) Methods: The effects of omeprazole, a proton pump inhibitor (PPI), on oxaliplatin-induced peripheral neuropathy (OIPN) was investigated using an in vivo model and a real-world database. (3) Results: In a rat model, oxaliplatin (4 mg/kg, i.p., twice a week for 4 weeks) caused mechanical hypersensitivity accompanied by sciatic nerve axonal degeneration and myelin sheath disorder. Repeated injection of omeprazole (5–20 mg/kg, i.p., five times per week for 4 weeks) ameliorated these behavioral and pathological abnormalities. Moreover, omeprazole did not affect the tumor growth inhibition of oxaliplatin in tumor bearing mice. Furthermore, clinical database analysis of the Food and Drug Administration Adverse Event Reporting System (FAERS) suggests that the group using omeprazole has a lower reporting rate of peripheral neuropathy of oxaliplatin-treated patients than the group not using (3.06% vs. 6.48%, *p* < 0.001, reporting odds ratio 0.44, 95% confidence interval 0.32–0.61). (4) Conclusions: These results show the preventing effect of omeprazole on OIPN.

## 1. Introduction

Oxaliplatin is indicated for the treatment of colorectal, pancreatic, and gastric cancers. In colorectal cancer, the development of regimens such as FOLFOX (combination of oxaliplatin, fluorouracil, and leucovorin), CapeOx (oxaliplatin and capecitabine), FOLFOXIRI (oxaliplatin, irinotecan, fluorouracil, and leucovorin), and SOX (oxaliplatin and S-1) has improved treatment outcomes [1,2,3,4,5,6]. However, it has a high risk of acute cold-induced neurotoxicity and a chronic cumulative neuropathy [7,8,9]. These peripheral neuropathies are dose-limiting toxicities and are often a barrier to continued treatment.

The effects of various drugs on peripheral neuropathy have been investigated in both basic and clinical studies [10,11,12,13,14]. However, according to the American Society of Clinical Oncology Guidelines for Chemotherapy-Induced Peripheral Neuropathy in 2020, there are no effective drugs for the prevention of peripheral neuropathy, and the only drug recommended for use in treatment is duloxetine [15]. Therefore, there is still a strong need to find effective drugs for prevention and treatment of peripheral neuropathy.

Omeprazole, a proton pump inhibitor (PPI), is used for the treatment of gastric or duodenal ulcers. Some reports show that omeprazole has an antioxidant effect [16,17]. There are many reports on the mechanism of oxaliplatin-induced peripheral neuropathy (OIPN) expression related to oxidative stress [18,19,20] so omeprazole could suppress OIPN via an antioxidative effect. In this study, we evaluate the effect of omeprazole on OIPN using an in vivo model and clinical database analysis using the Food and Drug Administration Adverse Event Reporting System (FAERS).

## 2. Results

### 2.1. Effect of Omeprazole on Mechanical Hypersensitivity Induced by Oxaliplatin in Rats

Repeated administration of oxaliplatin (4 mg/kg) caused a significant decrease in threshold at week 4 (Figure 1; Week 4, *p* < 0.01). Repeated doses of omeprazole (20 mg/kg) significantly reduced the threshold decrease induced by oxaliplatin (Figure 1; Week 4, *p* < 0.05). On the other hand, no significant difference was observed at the 5 mg/kg dose (*p* = 0.086), suggesting a dose-dependent neuropathic inhibitory effect of omeprazole.

### 2.2. Effect of Omeprazole on Axonal Degeneration and Myelin Sheath Disorder of Sciatic Nerves Induced by Oxaliplatin in Rats

Figure 2A shows a cross-sectional image of rat sciatic nerve tissue stained with toluidine blue, in which the blue stained area is the myelin sheath and the white area inside it is the axon. Each axon was evaluated using circularity. Significant axonal degeneration was observed in the oxaliplatin of the 5% glucose solution group (*p* < 0.01; Figure 2B) and the suppression of axonal degeneration was observed in the omeprazole combination group (Figure 2B). Myelopathy was evaluated using g-ratio and myelin sheath thickness. There was significant myelopathy in the oxaliplatin of the 5% glucose solution group (*p* < 0.01; Figure 2C,D) and significant suppression of myelopathy in the omeprazole combination group (*p* < 0.01; Figure 2C,D).

### 2.3. Effect of Omeprazole on the Anti-Tumor Effect of Oxaliplatin in Tumor-Bearing Mice

The oxaliplatin treatment (6 mg/kg, i.p.) significantly inhibited the increase in tumor volumes compared with the vehicle treatment (*p* < 0.01; Figure 3). Omeprazole had no significant effect on tumor growth inhibition in conjunction with oxaliplatin treatment (Figure 3).

### 2.4. Effects of PPIs on Reporting Ratio of Peripheral Neuropathy in Oxaliplatin-Treated Patients in FAERS

Of the 49,352 adverse event reports for patients using oxaliplatin in the FAERS database, 3153 (6.39%) reports included peripheral neuropathy in oxaliplatin-treated patients. The reported incidence of peripheral neuropathy in oxaliplatin-treated patients was significantly lower when pantoprazole, omeprazole, and rabeprazole were used concomitantly (Figure 4). Reporting odds ratios (RORs) of peripheral neuropathy in oxaliplatin-treated patients were 0.54 [95% confidence interval (CI) 0.41–0.72, *p* < 0.001] for pantoprazole, 0.45 [95% CI 0.33–0.62, *p* < 0.001] for omeprazole, 0.31 [95% CI 0.12–0.84, *p* < 0.05] for rabeprazole. In addition, there was a significantly lower rate of peripheral neuropathy in oxaliplatin-treated patients reported in patients using any PPI concomitantly (ROR [95%CI] = 0.66 [0.57–0.77], *p* < 0.001).

## 3. Discussion

In the rat model oxaliplatin-induced mechanical hypersensitivity was significantly suppressed by omeprazole treatment. In the sciatic nerve tissue there was significant suppression of axonal injury and myelin sheathing disorder. These results suggest that omeprazole may suppress the development of peripheral neuropathy by inhibiting neurodegeneration, especially myelopathy and axonopathy.

The mechanism by which omeprazole suppresses myelopathy is not clear, but there have been several related reports. The first is the involvement of neuregulin 1 (NRG1), a myelinating factor. It has been reported that NRG1 is decreased in oxaliplatin-treated rats [21]. The second is the involvement of the ERK/MAPK pathway in myelin sheath remodeling. Omeprazole activates ERK/MAPK pathway which differentiates Schwann cells and improves demyelinating symptoms [22]. The third is the involvement of uptake into the dorsal root ganglia (DRG) of the spinal cord. The development of oxaliplatin-induced neuropathy involves uptake into the DRG and subsequent neuronal cell body damage [23]. The organic cation transporter 2 (OCT2) is involved in this uptake into the DRG. Since omeprazole has been reported to decrease the expression of OCT2 in the kidney [16], it may similarly inhibit OCT2-mediated uptake into the DRG and the development of neuropathy. Finally, there is the involvement of oxidative stress. There are several reports that oxidative stress is involved in the expression of OIPN [18,19,20]. Dimethyl fumarate, a drug for demyelinating diseases, has been reported to contribute to the improvement of demyelinating symptoms through its antioxidant effect [24] and to have an OIPN inhibitory effect via attenuation of axonopathy [25]. Since omeprazole also has antioxidant properties, it may reduce oxidative stress and inhibit myelopathy and axonopathy by a similar pathway. Its antioxidant effects are common to many PPIs [17], and a correlation may exist between its differential antioxidant capacity and its neuropathy-inhibiting effects. Further research is needed to elucidate the mechanism of suppression of neurodegeneration.

Large-scale health information databases are beginning to be used in drug discovery and development. More than 3 million spontaneous reports of adverse events have been registered in FAERS, making it an effective tool for comprehensive risk assessment of adverse drug events and drug repositioning studies [26,27,28,29]. While this database has the weakness that it does not contain information on all patients who have used a particular drug, but only those who have reported adverse events, it has the strength of being able to analyze a large number of cases in clinical practice. Our study indicated that the ROR of peripheral neuropathy in oxaliplatin-treated patients was significantly decreased in patients concomitantly administrated with pantoprazole, omeprazole, and rabeprazole. There was also a significant decrease in ROR for any PPI. The reason for the significant suppression by PPIs as a whole may be due to the heavy influence of omeprazole and pantoprazole, which are reported in large numbers. However, the possibility that the suppression of peripheral neuropathy ROR is a class effect of PPIs cannot be denied, so this should be clarified in future studies. Confounding factors should also be considered; however, it is impossible to obtain information on patient backgrounds from the database and only indirect inferences can be made. For example, it is assumed that PPIs may be used in combination with NSAIDs due to the gastrointestinal side effects of NSAIDs. In fact, there were only 137 cases (0.27%) of omeprazole combined with NSAIDs (loxoprofen, ibuprofen, or diclofenac) in the database, and NSAIDs themselves have no inhibitory effect on peripheral neuropathy (ROR = 1.27; 95% CI [0.97–1.67]) (result not shown). It is very difficult to examine these examples one by one from the database. In order to solve this problem, we are planning to conduct actual retrospective clinical studies including patient backgrounds.

While many drugs have been reported in basic research as having the potential to inhibit the neuropathy caused by oxaliplatin, few drugs have developed sufficient evidence in clinical studies [14]. We have also shown the effects of various drugs, such as sodium channel blockers, calcium channel blockers, NMDA receptor antagonists, amitriptyline, exenatide, riluzole, alogliptin, dimethyl fumarate, donepezil, and goshajinkigan, on OIPN in animal models [14,24]; however, only a few of these drugs have been studied in clinical practice [12,30,31,32]. Many problems, such as the cost and time required to conduct clinical research and the consideration of safety in clinical application, are considered to be the reasons why the results of basic research do not translate into clinical applications. Retrospective studies can be helpful as a first bridging step from basic research to clinical research. Since PPIs are already frequently used drugs for gastrointestinal symptoms, it will be easy to conduct a retrospective research study. Moreover, our animal study indicated that the antitumor effect of oxaliplatin was also evaluated using a tumor-bearing mice model, and omeprazole was found to have no effect on it. Furthermore, it is a known clinical fact that omeprazole is a safe drug with few adverse events. Therefore, it is possible that PPIs may be clinically applied as a prophylactic agent for OIPN in the future. It is expected that clinical evidence on the usefulness of PPIs including omeprazole for OIPN will be developed.

## 4. Materials and Methods

### 4.1. Animals

We used male Sprague Dawley rats (six-week-old, 200–250 g, Japan SLC, Inc., Shizuoka, Japan) for the peripheral neuropathy model, and BALB/c mice (six-week-old, 15–25 g, Japan SLC, Inc.) for the tumor-bearing model. Animals were kept at constant temperature and humidity and under a 12 h light-dark cycle (light period: 7:00–19:00). Solid samples and water were allowed to be consumed freely. Animal experiments were conducted in accordance with the Kyushu University Animal Experiment Regulations, related laws and regulations, and ARRIVE guidelines 2.0 (Animal Research: Reporting of In Vivo Experiments), and with the approval of the Kyushu University Animal Experiment Committee.

### 4.2. Drugs

In the oxaliplatin-induced peripheral neuropathy rat model, oxaliplatin (4 mg/kg, Yakult Honsha Co., Ltd., Tokyo, Japan) or a vehicle (5% glucose solution) was injected intraperitoneally (i.p.) twice a week for four weeks. Omeprazole (20 mg/kg, FUJIFILM Wako Pure Chemical Corporation, Osaka, Japan) was injected intraperitoneally (i.p.) five times a week for four weeks. In the tumor-bearing mice model, oxaliplatin (6 mg/kg) was administered intraperitoneally twice a week for 2 consecutive weeks, and omeprazole (30 mg/kg) was administered intraperitoneally five times a week. The doses of these drugs were determined based on previous reports [16,33,34,35].

### 4.3. Von Frey Test for Mechanical Hypersensitivity

This test was performed before (day 0) and on day, 7, 14, 21, and 28 after the initiation of drug administration. The rats were placed on a wire mesh for 30 min prior to the test, and after sufficient acclimatization, the rat hind paws were stimulated from under the mesh for 6 s at a time using von Frey filaments (Aesthesio^®^; DanMic Global. LLC, San Jose, CA, USA). The up–down method was used for the measurement, and the filament intensity that caused the escape response was recorded as the escape response threshold.

### 4.4. Assessment of Sciatic Nerve Axonal Degeneration and Myelin Sheath Disorder

Sciatic nerves were harvested from rats 4 weeks after the initiation of drug administration and fixed in 0.1 M phosphate buffer containing 2% glutaraldehyde (pH 7.4, 4 °C) for 4 h. After replacement with 8% sucrose solution and Epon inclusion, thin sections were made and stained with toluidine blue. Each section was evaluated using a fluorescence microscope (BX51; Olympus Co., Tokyo, Japan). The axon circularity was calculated as a quantitative index of axonal degeneration [36]. The *g*-ratio and myelin sheath thickness were calculated as quantitative indicators of myelin sheath damage [21].

### 4.5. Assessment of Tumor Growth in Tumor-Bearing Mice

C26 cells (mice colorectal cancer cells) were used to prepare a tumor-bearing animal model. Cells were obtained from the Cell Resource Center for Biomedical Research (Tohoku University, Miyagi, Japan). C26 cells were cultured in RPMI 1640 medium (Sigma–Aldrich Co. LLC., St. Louis, MO, USA) containing 2 mM L-glutamine, 100 units/mL penicillin, 100 μg/mL streptomycin, and 10% fetal bovine serum at 37 °C and 5% CO_2_. Cultured C26 cells (1.5×10^6^ cells) were transplanted subcutaneously into the right hind paw of BALB/c mice. From day 3 to day 14 after C26 transfer, drugs were administered. The measurements of tumor volumes were performed on days 3, 5, 7, 10, 12, and 14. Tumor volume was calculated by measuring the length, width, and thickness of the hind paws of mice on both sides and approximating the difference between them as shown in Equation (1):(1)$$\mathrm{volume}\;(\mathrm{mm}3)\;=\;\frac{\pi }{6}\;\times \;\mathrm{thickness}\;(\mathrm{mm})\;\times \;\mathrm{length}\;(\mathrm{mm})\;\times \;\mathrm{width}\;(\mathrm{mm})$$

### 4.6. Analysis of FAERS Data

To search for candidate drugs to suppress OIPN, we extracted data using CzeekV Pro (Version 5.0.32, Intage Healthcare Inc., Tokyo, Japan, accessed April 2021) for FAERS reported data from 2004 to 2020. Of the 13,829,818 adverse events registered during the study period, 49,352 adverse event reports from patients using oxaliplatin were included in the study. We calculated the number of reported OIPNs, the reporting rate, and the Reporting Odds Ratio (ROR) for PPI in the group with and without the drug. OIPN was defined as reports of peripheral neuropathy, peripheral sensory neuropathy, or peripheral sensorimotor neuropathy in patients using oxaliplatin.

The ROR and its 95% Confidence Interval (CI) were calculated from Equations (2) and (3):(2)$$\mathrm{ROR}=\frac{n11/n21}{n12/n22}$$
(3)$$95\%\mathrm{CI}=\exp\left[\log\left(\mathrm{ROR}\right)\pm 1.96\sqrt{\frac{1}{n11}+\frac{1}{n12}+\frac{1}{n21}+\frac{1}{n22}\;}\;\right]$$

In the above equation, n11 refers to patients who used concomitant medications and reported OIPN, and n12 refers to patients who used concomitant medications and did not report OIPN. n21 refers to patients who did not use concomitant medications and reported OIPN. n22 refers to patients who did not use concomitant medications and did not report OIPN.

### 4.7. Sample Sizes

Each sample size was calculated by the following equation: n = 2σ^2^/δ^2^ × (Z_1−α/2_ + Z_1−β_)^2^ (n, minimum sample size; α, the probability of type I error, 5%; β, the probability of type II error, 20%; σ, variance; δ, the difference between two groups). The minimum sample size in the von Frey test was calculated as eight, using σ and δ values resulting from the preliminary experiments (σ = 5.5 g, δ = 8.0 g). In the same way, the minimum sample sizes of experiments of tumor-bearing mice were determined as eight (σ = 145 mm^3^, δ = 210 mm^3^).

### 4.8. Statistical Analysis

The results are expressed as the mean with standard error of the mean. JMP14 software (SAS Institute Inc., Cary, NC, USA) was used for statistical analysis. In the study using the rat model, comparisons between multiple groups were made using a one-way ANOVA followed by the Tukey–Kramer test. In the FAERS analysis, each dataset was tested by Chi-square test. If the number of examples for each item was less than five, the Yates correction was applied. A risk rate of less than 5% (*p* < 0.05) was considered a significant difference.

## 5. Conclusions

In conclusion, our results suggest that repeated administration of omeprazole suppresses OIPN without affecting the antitumor effect of oxaliplatin. The preventive effect of omeprazole on OIPN was also supported by the result obtained in the analysis of the database, and the data can be said to reflect clinical practice. Therefore, omeprazole may be an effective prophylactic agent for oxaliplatin-induced peripheral neuropathy.

## Figures and Tables

**Figure 1 ijms-23-08859-f001:**
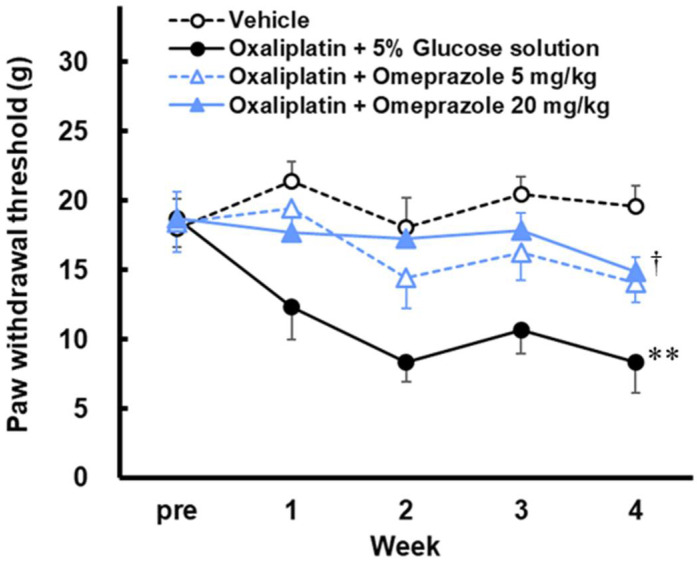
Effect of omeprazole on mechanical hypersensitivity induced by oxaliplatin in rats. Oxaliplatin (4 mg/kg) was injected intraperitoneally twice a week for four weeks. Omeprazole (5 and 20 mg/kg) was injected intraperitoneally 5 times a week for four weeks. The von Frey test was performed before the first drug administration (week 0) and once a week. Thresholds are expressed as the mean with S.E.M. (n = 8–9), ** *p* < 0.01 vs. Vehicle, † *p* < 0.05 vs. Oxaliplatin with 5% glucose solution, One-way ANOVA followed by Tukey–Kramer test.

**Figure 2 ijms-23-08859-f002:**
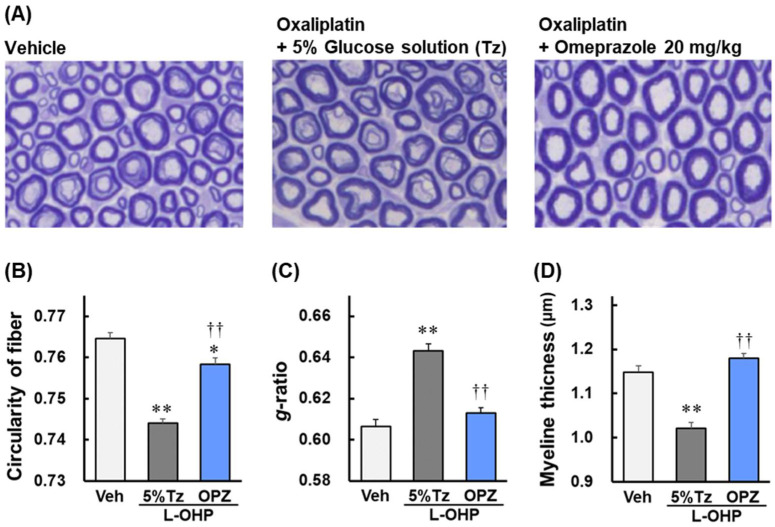
Effect of omeprazole (OPZ) on axonal degeneration and myelin sheath disorder of sciatic nerves induced by oxaliplatin (L-OHP) in rats. On day 28, sciatic nerves were harvested and stained with toluidine blue. The images (**A**) are magnified 40×. The circularity of fiber (**B**), *g*-ratio (**C**), and myelin thickness (**D**) were analyzed using ImageJ 1.53 software. These results are expressed as the mean with S.E.M. (n = 4–6), * *p* < 0.05, ** *p* < 0.01 vs. Vehicle, †† *p* < 0.01 vs. Oxaliplatin with 5% glucose solution, One-way ANOVA followed by Tukey–Kramer test.

**Figure 3 ijms-23-08859-f003:**
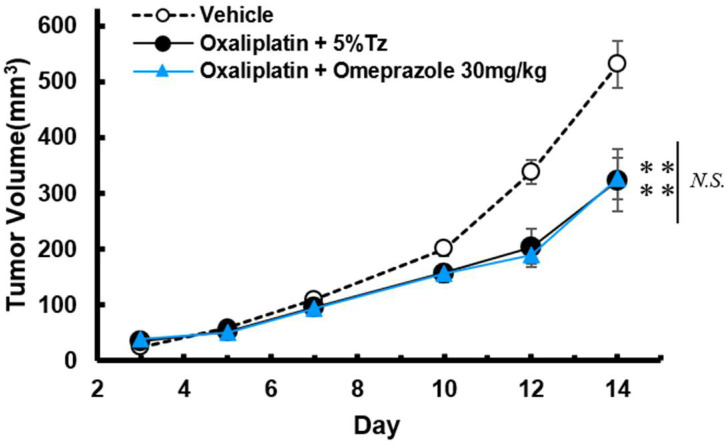
Effect of omeprazole on the anti-tumor effect of oxaliplatin in tumor-bearing mice. Cultured C26 cells (1.5 × 10^6^ cells) were transplanted subcutaneously into the right sole of BALB/c mice. Oxaliplatin (6 mg/kg) was injected intraperitoneally twice a week for two weeks. Omeprazole (30 mg/kg) was injected intraperitoneally 5 times a week for two weeks. Tumor volume was measured every 2 or 3 days from day 3 to day 14 after C26 transfer. Tumor volume was expressed as the mean with S.E.M. (n = 8), ** *p* < 0.01 vs. Vehicle, One-way ANOVA followed by Tukey–Kramer test.

**Figure 4 ijms-23-08859-f004:**
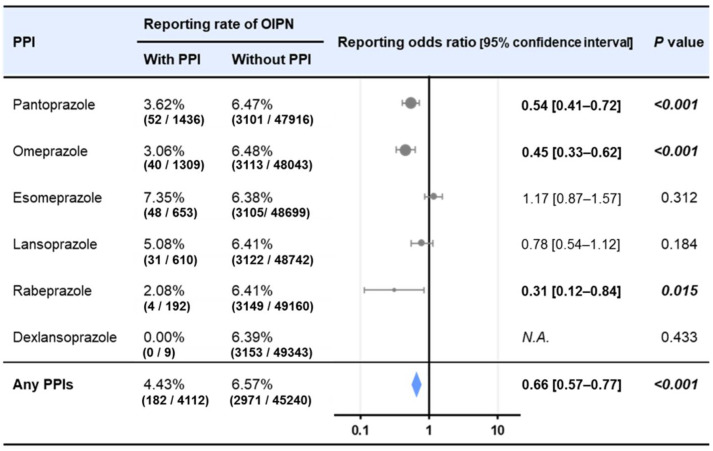
Effects of proton pump inhibitors (PPIs) on the reporting ratio of peripheral neuropathy in oxaliplatin-treated patients in the Food and Drug Administration Adverse Event Reporting System (FAERS). The report data were extracted using CzeekV Pro (version 5.0.23, INTAGE Healthcare Inc., Tokyo, Japan, accessed April 2021). A total of 49,352 adverse event reports from patients using oxaliplatin were included in this study, peripheral neuropathy in oxaliplatin-treated patients was defined as reports of peripheral neuropathy, peripheral sensory neuropathy, or peripheral sensorimotor neuropathy in patients using oxaliplatin.

## Data Availability

The data that support the findings of this study are available from the corresponding author upon reasonable request.

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
