# Peer review of "Omeprazole Suppresses Oxaliplatin-Induced Peripheral Neuropathy in a Rodent Model and Clinical Database"

_ijms, 2022, doi:10.3390/ijms23168859_

Round 1
Reviewer 1 Report
In this study, the authors have shown that omeprazole, a proton pump inhibitor, suppressed the oxaliplatin-induced mechanical allodynia and axonopathy/myelopathy in rat model, which was consistent with the data from FAERS database. Overall, the topic addressing the efficacy of PPIs on OIPN is interesting, and the results are clearly presented. However, there are numerous points that should be addressed by the authors, as described below.
1) As the authors themselves pointed out, the major weak point of this study is that the mechanisms of the inhibitory effect of omeprazole are unknown. The authors have shown that other PPIs, pantoprazole and rabeprazole also reduced the reported incidence of OIPN in FAERS analysis. The authors should also show the effects of these PPIs to demonstrate the class effect of PPIs. In addition, the authors discussed the involvement of the antioxidant effect of omeprazole in the inhibitory effect. Thus, the authors should discuss the involvement of the antioxidant effect of other PPIs.
2) For the description of the results, the authors should avoid to use the term "mechanical allodynia" for the decreased paw withdrawal threshold to von Frey filament stimulation in experimental animal models. "Mechanical allodynia" should be changed by "mechanical hypersensitivity" as the reflexive test does not allow to truly assess allodynia in animals.
3) In page 2, line 48-49, the authors described “the only drug approved for use in treatment is duloxetine”, but duloxetine is not approved for CIPN. The authors should change f to “the only drug recommended for use in treatment is duloxetine”.
4) In Results section, the authors should describe the results about the dose-dependent effect of omeprazole, as the drug exerted inhibitory effect at 20 mg/kg, but not 5 mg/kg in Figure 1.
5) The authors should avoid to use the term "moontherapy" when referring to administration of oxaliplatin with 5% glucose solution. "Combination therapy" is therapy that uses more than one medication to treat a disease. Oxaliplatin is an anti-cancer agent, but omeprazole is not used as an anti-cancer agent, but as a candidate drug to prevent OIPN.
6) In page 4, line 96-97: “Tumor volume was measured every 2 or 3 days from day 3 to day 14 after C26 transfer”. In Figure 3, only the data from day 3 to day 12 after C26 transfer are shown, and the data on day 14 is not included.
7) In page 4, line 101: “3,153 (6.39%) were OIPN reports”. Is this explanation correct? “3,153 (6.39%) reports included OIPN”?
Reviewer 2 Report
Interesting work by Mine et al. on the preventive effects of omeprazole on the oxaliplatin related-CIPN.
I have no comment on the animal part, which is quite clear and well constructed.
Whereas, I have some concerned on the FAERS analysis. The results are crystal clear, however, there are many confounding factors which are not identified and may influence the results. We know nothing on the profile of patients in this database. For example, it is possible that many patients taking PPI used also analgesic drugs such as non-steroidal anti-inflammatory drugs, which can lead to digestive adverse effect (ulcer), and lead to PPI treatment. So I would recommend to deeply discuss all the possible biases/ confounding factors that can be associated to this type of analysis.
In the same way, it would be better to talk about oxaliplatin-treated patients and reporting peripheral neuropathy, instead of OIPN. We don’t know if the patients take other neurotoxic drugs or not.
So the interpretation of the FAERS analysis, which is very interesting, should be make with caution.
In the discussion part, second paragraph, “oxaliplatin suppresses myelopathy”, is it really oxaliplatin or omeprazole? Because I understand that it is omeprazole.
The authors should discuss more deeply the potential mechanism of omeprazole. Because beside to potential antioxidative effects, it is possible that omeprazole modified the PK of oxaliplatin. PPI and omeprazole are known to modulate membrane transporters such as SLC22a ones, which also mediate platinum derivatives influx into cells (and neurotoxicity). This point could be also discussed.
In the last paragraph of the discussion, we have the feeling that the authors cite many works from their lab, which are not strictly relevant of this topic. (self citation…?).
Round 2
Reviewer 1 Report
The authors have appropriately responded to my concerns.
This manuscript is a resubmission of an earlier submission. The following is a list of the peer review reports and author responses from that submission.
Round 1
Reviewer 1 Report
Dear Authors,
Thank you for submitting your work.
Please see below some specific comments:
- Please add details regarding the sample size calculation and how you came up with that number of animals. This would be very important considering that this study included a limited number of rodents.
- Please add some more details regarding von Frey filaments test (e.g.: did you start with the smallest filament? Did you apply a selection of filaments or could you use potentially all 20 filaments in case of no response in a specific animal?). This would be important as allodynia is difficult to test in animals, and to distinguish from nociception/acute somatic pain or simply escaping behaviour.
Best wishes.
Reviewer 2 Report
In this manuscript, the authors showed evidence of pump inhibitiors such as omeprazole as a new strategy to treat chemotherapy induced peripheral neuropathy. The carried out behavioral studies in mice, stained peripheral nerves to demonstrate loss in myelin and degeneration, and evaluated a database of side effects reported by patients as a means to support the preclinical evidence. Finally, the effect of omeprazole upon tumor growth was studied, and none effect was observed.
The paper is well written (at least for this non-native reviewer), and the methods, results and discussion sectionas are clearly presented. However, despite the results are promising, the data are still very preliminary to be published as is. The authors should explore more mechanistic evidence supporting the effects of omeprazol as follows:
i) What is the effect of omeprazol upon other nociceptive behavior (cold and heat sensitivity) elicited by oxaliplatin?
ii) what is the role of oxidative stress and neuroinflamation on omeprazol-mediated control of neurophaty? We do now have evidence that oxaliplatin causes oxidative stress in nociceptive pathway, and that activation of glial cells in dorsal horn, and macrophages in peripheral nerves also contribute to neurophaty phenotype; so authors could build upon these previous evidence..
iii) should omeprazol impact the activation of nociceptive pathways, such as TRPA1, CGRP and other involved in nociceptive pathways?